# Size Effect of Electrical and Optical Properties in Cr^2+^:ZnSe Nanowires

**DOI:** 10.3390/nano13020369

**Published:** 2023-01-16

**Authors:** Yuqin Zhang, Shi He, Honghong Yao, Hao Zuo, Shuang Liu, Chao Yang, Guoying Feng

**Affiliations:** 1College of Mathematics and Physics, Chengdu University of Technology, No.1 East Third Road, Erxianqiao, Chenghua District, Chengdu 610059, China; 2Institute of Laser & Micro/Nano Engineering, College of Electronics & Information Engineering, Sichuan University, No.24 South Section 1, Yihuan Road, Chengdu 610064, China

**Keywords:** Cr^2+^:ZnSe nanowires, size effect, first-principles, quantum confinement effects, quantum size effects

## Abstract

Previous studies have shown that the nano-crystallization process has an appreciable impact on the luminescence properties of nanocrystals, which determines their defect state composition, size and morphology. This project aims to explore the influence of nanocrystal size on the electrical and optical properties of Cr^2+^:ZnSe nanowires. A first-principles study of Cr^2+^:ZnSe nanowires with different sizes was carried out at 0 K in the density functional framework. The Cr^2+^ ion was found to prefer to reside at the surface of ZnSe nanowires. As the size of the nanocrystals decreased, a considerable short-wave-length shift in the absorption of the vis-near infrared wavelength was observed. A quantum mechanism for the wavelength tunability was discussed.

## 1. Introduction

Mid-infrared lasers play a critical role in the areas of science, economy and national defense because they are widely used in molecular spectroscopy, non-invasive medical diagnostics, the processing of polymers and many other areas [1,2]. TM-doped II–VI compounds stand out among other laser crystals as they have a wide bandgap and possess several important features [3]. Cr^2+^:ZnSe attracts attention due to its outstanding properties, such as low optical phonon cutoff, large absorption and emission cross-section, and high quantum efficiency [4]. After recent decades of development, continuous (CW) and pulsed laser technologies based on Cr^2+^:ZnSe have matured [5,6,7,8,9,10], and entered practical applications [11]. To date, works aiming to achieve miniaturization, integration, portability, broad tunability, a high peak and average power micro-nano lasers are of the greatest interest [12,13]. Although successful experiments have been reported for the emission of TM^2+^-doped II–VI nanoparticles [10,11], few have been performed in the mid-IR spectral range [12,13]. Studying the properties of TM^2+^-doped II–VI nanocrystals is of great importance. First-principles calculations make it easier to obtain the properties of nanomaterials and provide a theoretical basis for the design of nanomaterials [14,15,16].

However, due to nanocrystals’ self-cleaning effect, the solubility of dopant ions in nanocrystals is much lower than that of bulk materials. Even if dopant ions are successfully introduced into nanocrystals, it is difficult to determine the location of dopant ions in nanocrystals using existing characterization methods, and only a few analytical techniques can distinguish whether impurities are present on the surface of or inside the material. For example, for the Mn^2+^ ion, the spin interaction shown by the electron paramagnetic resonance technique can distinguish the local environment of the impurity ion [17,18]. For Co^2+^ ions, the absorption spectrum is very sensitive to the local environment of the impurity ions and can be used to detect the position of the ions [19,20]; magnetic circular dichroism spectroscopy is also an effective detection method [21,22]. First-principles calculation provides another way to acquire information on the material’s position.

In the present work, using a first-principles calculation method, we report a fundamental study on the size effect of electrical and optical properties of Cr^2+^:ZnSe nanowires (NWs). Our analysis is mainly focused on the size effect on the energy level splitting of the Cr(d) orbitals. Through the calculation of the band structure, density of state (DOS) and absorption spectra of Cr^2+^:ZnSe NWs in different sizes, the change in material property law that occurs with size is obtained, and the intrinsic size regulation mechanism is revealed. By calculating the formation energies of Cr^2+^:ZnSe NWs at different doping positions, the positions of Cr^2+^ ions in the nanowire is predicted. This investigation will provide some new insights into the design and synthesis of Cr^2+^:ZnSe nanocrystals and lay the foundation for the realization of new photoelectronic devices. 

## 2. Materials and Methods

Bulk ZnSe presents a cubic zinc-blende structure at low temperatures. During the doping and nanocrystallization process, this zinc-blende-structure can be retained. Many similar experiments have been reported. Here, sphalerite ZnSe NWs, alone [001] and with a square cross-section of a different diameter, were considered in this study: NW-1(8.5 Å), NW-2(14.2 Å), NW-3(19.8 Å), as shown in Figure 1. The ZnSe NWs models were designed by cutting out a fragment of 7 × 7 × 1 bulk ZnSe crystals. A 20 Å vacuum space was constructed to minimize the artificial interaction between neighboring NWs due to the periodic boundary conditions. The NWs’ surface Zn and Se bonds were saturated with H1.5 and H0.5 [23]. The Cr^2+^:ZnSe NWs were structured by replacing a Zn atom with a Cr atom in the unit cell of the pristine NWs models, resulting in Cr^2+^ concentrations of 12.5%, 5.6% and 3.1%, respectively. The chemical formulas for NW-1, NW-2 and NW-3 are H_16_Cr_1_Zn_7_Se_8_, H_24_Cr_1_Zn_17_Se_18_ and H_32_Cr_1_Zn_31_Se_32_, respectively. In each ZnSe NW with different diameters, we chose nonequivalent Zn positions from the center to surface, labeled as 1, 2, 3, 4 (as shown in Figure 1), to study the effect of different doping sites. Thus, we obtained nine models: NW-1(site 1), NW-1(site 2), NW-2(site 1), NW-2(site 2), NW-2(site 3), NW-3(site 1), NW-3(site 2), NW-3(site 3), NW-3(site 4).

The optimization of Cr^2+^:ZnSe NWs, and the calculation of total energies, electrical and optical properties, were carried out with the Vienna Ab-initio simulation package (VASP, version 5.2) [24,25] based on density functional theory (DFT). Generalized Gradient Approximation (GGA) in Perdew–Burke–Ernzerhof (PBE) [26] was used to describe the exchange and correlation functional, and Projector Augmented Wave (PAW) [27] potential was used to describe the ion–electron interaction. PBE functional calculations generally underestimate the band gap value of semiconductor materials, while the meta-GGA and HSE functions can effectively improve this defect. The underestimation of the electric bandgap from DFT calculations also derives from the difference in the temperature in the DFT calculations (0 K) and the experimental conditions. The excitonic effect is another important aspect, which should be considered to improve the drawback [28]. In this work, we focused on the tendency of the electrical and optical properties in different-size Cr^2+^:ZnSe NWs, so the cost-effective PBE functional was applied in the calculations. The valence electron configurations considered in the calculations were Se (3s^2^3p^4^), Zn (3d^10^4s^2^) and Cr (3d^5^4s^1^), respectively. The electron wave functions were expanded as a plane wave with a cutoff energy of 500 eV. The Brillouin zone was sampled with 1 × 1 × 4 Monkhorst–Pack k-points. A higher cutoff energy and a denser k points mesh were adopted to examine the accuracy of the calculations, the results showing almost no change in the energy and geometry structure. To optimize the geometry, the ion position, cell volume and shape were allowed to change, the force on each atom was converged to less than 1 × 10^−3^ eV/Å, the maximum displacement for each ion was set as 0.001 Å, and the total energy of the system was converged to lower than 1 × 10^−4^ eV/atom. Based on the optimized structures, the total energy, band structure, density of state and optical properties of Cr^2+^:ZnSe NWs were calculated. 

## 3. Results

### 3.1. System Stability

Defect formation energy is an important parameter that reflects the ease of formation of specific defects and the stability of the defect systems. By calculating and analyzing the formation energy of Cr^2+^:ZnSe NWs with Cr^2+^ at different doping sites, the positions of Cr^2+^ ions in the nanowires can be determined. The defect formation energy is usually calculated as:(1)$$E_{form}=E_{defect}-E_{pure}-\overset{Nspecies}{\underset{s=1}{\sum }}n_{s}\mu_{s}$$
where $E_{defect}$ and $E_{pure}$ represent the total energy of a supercell with and without the defect; $n_{s}$ is the number of atoms of type s that were added ($n_{s}$>0) or removed ($n_{s}$<0) to create the defect; the chemical potential of atomic species is given as s; and the sum of elemental species is Nspecies. Here, one Zn atom was replaced by a Cr atom; the defect formation energy can be expressed as:(2)$$E{\left(\mathrm{NW}\right)}_{form}=E{\left(\mathrm{NW}\right)}_{Cr^{2+}:ZnSe}-E{\left(\mathrm{NW}\right)}_{ZnSe}-\left(\mu_{Cr}-\mu_{Zn}\right)$$
where $E{\left(\mathrm{NW}\right)}_{Cr^{2+}:ZnSe}$ and $E{\left(\mathrm{NW}\right)}_{ZnSe}$ are the total energy of Cr^2+^:ZnSe NWs and ZnSe NWs, and $\mu_{Cr}$ and $\mu_{Zn}$ are the chemical potential of Cr and Zn atoms, respectively. By calculating the energy difference in CrSe NWs and ZnSe NWs, the chemical potential $\mu_{Cr}-\mu_{Zn}$ of replacing one Zn atom with a Cr atom in NWs can be obtained, and its value is −7.11 eV.

The defect formation energy of Cr^2+^:ZnSe NWs is listed in Table 1. Note that the defect formation energies are all positive, suggesting that the doped models have a lower stability owing to their higher energy. Extra energy is needed when substituting the Cr atom with Zn atom in ZnSe NWs, which coincides with the preparation process of nanocrystals. It can be concluded that the formation energy increases as the diameter of NWs decreases, indicating that NWs with a larger diameter are easier to dope. In addition, for NWs with the same diameter, the defect formation energy at the surface of the NWs is smaller than that inside, indicating that Cr^2+^ prefers to reside at the surface of ZnSe NWs. This result is consistent with the conclusion of B-doped Si NWs [29], the solubility of Cr^2+^ on the ZnSe NWs surface is higher than that inside. And this may relate to the self-cleaning effect of nanocrystals. To study the size effect in Cr^2+^:ZnSe NWs, the most stable models, NW-1(Site 2), NW-2(Site 3) and NW-3(Site 3), were chosen as the research objects. The changes in the electronic structure and optical properties of Cr^2+^:ZnSe NWs with NW size were analyzed.

### 3.2. Electrical Properties

To analyze the modifications of the doping effect on the electrical properties, both the band structure and DOS are calculated based on the optimized structure. 

Figure 2 shows the band structure of Cr^2+^:ZnSe NWs. The valence band maximum (VBM) and the conduction band minimum (CBM) are located at the same ***k*** point Γ, indicating that the Cr^2+^:ZnSe NWs remain as direct bandgap semiconductors. It should be mentioned that, compared with pure ZnSe NWs, impurity bands (red lines) that lie around the Fermi Level are introduced into the bandgap after Cr atom doping, suggesting that more possibilities for electron transitions can be created. Table 2 shows the bandgap of Cr^2+^:ZnSe NWs and bulk. Compared to bulk Cr^2+^:ZnSe [30], the bandgaps of Cr^2+^:ZnSe NWs become larger and present a distinct size effect. In the bulk Cr^2+^:ZnSe, Van der Waals interactions exist between neighboring Cr^2+^:ZnSe bundles. When the vacuum space is intercalated into NWs, the electron motion is confined to the NW growth direction, meaning that the spatial confinement of electron and hole within the Cr^2+^:ZnSe NWs plays an essential role in the electronic structures.

The effective mass of an electron ($m_{e}^{*}$) or hole ($m_{h}^{*}$) determines its dynamics near the band minimum or band maximum of high symmetry in the ***k*** space. In general, the effective mass tensor **M** with components is defined as [31]:(3)$$\frac{1}{M\left(k\right)ij}=\left(\pm \frac{1}{\hbar^{2}}\right)\frac{\partial^{2}E}{\partial_{k_{i}}\partial_{k_{j}}}\left(k\right),i,j\in \left\{1,2,3\right\}$$
where + or − depend on whether ***k*** is near the band minimum (electrons) or the band maximum (holes), respectively. The effective masses of electron and hole at CBM and VBM for Cr^2+^:ZnSe bulk and NWs can be obtained by the numerical calculation of the second derivatives at the CBM and VBM in the band structure, with $m_{e}$ representing the electron rest mass, as given in Table 2. The electrons effective masses are significantly larger than the holes for Cr^2+^:ZnSe NWs, indicating that Cr^2+^:ZnSe NWs belongs to the class of semiconductors with light excited holes and heavy excited electrons. In general, the carries transfer rate is inversely proportional to the effective masses of the carries. The calculated results demonstrate that the holes move much faster than their corresponding electrons in Cr^2+^:ZnSe NWs. While, the situation is opposite for Cr^2+^:ZnSe bulk, electrons effective masses are significantly lighter than the holes.

DOS is considered having a better understanding about the electronic behavior of the doping system. As can be observed from Figure 3, the impurity bands emerge in the bandgap are mainly composed of Cr(d) orbital electrons. The impurity bands span the Fermi level and are divided into electronically occupied bands and empty bands. The energy difference between the electron-occupied bands and empty bands increases with the decrease in NWs diameter, and is larger than the bulk Cr^2+^:ZnSe [32].

To further analyze the splitting characteristics of the impurity orbitals, the partial density of state (PDOS) of Cr(d) orbitals is calculated and shown in Figure 4. The Cr(d) orbitals are classified into triply degenerated $t_{2}$ orbitals ($d_{xy}$, $d_{yz}$, $d_{zx}$) and doubly degenerated e orbitals ($d_{x^{2}-y^{2}}$, $d_{z^{2}}$), which lie above and below the Fermi level, the same as in the bulk case. With the decrease in Cr^2+^:ZnSe NWs diameter, the positions of the electron-occupied orbitals did not significantly change, but the electron-unoccupied orbitals moved in the high-energy direction, resulting in an increase in the splitting energy between e and $t_{2}$ orbitals.

### 3.3. Optical Properties

Investigating the optical material properties is crucial to understanding its optoelectronic properties. The frequency-dependent dielectric function $\varepsilon \left(\omega \right)$, which can be expressed as $\varepsilon \left(\omega \right)=\varepsilon_{1}\left(\omega \right)+\varepsilon_{2}\left(\omega \right)$, is a complex quantity that deals with the most important aspects of the optical properties. The imaginary part $\varepsilon_{2}\left(\omega \right)$ could be calculated from the momentum matrix elements between the occupied and unoccupied wave functions, and the real part $\varepsilon_{1}\left(\omega \right)$ can be further derived from $\varepsilon_{2}\left(\omega \right)$ using the Kramer–Kronig relationship [33]. With the knowledge of the dielectric function $\varepsilon \left(\omega \right)$, all the optical parameters could be obtained. The optical absorption coefficient $\alpha \left(\omega \right)$ could be obtained as:(4)$$\alpha \left(\omega \right)=\sqrt{2}\omega {\left[\sqrt{\varepsilon_{1}{\left(\omega \right)}^{2}+\varepsilon_{2}{\left(\omega \right)}^{2}}-\varepsilon_{1}\left(\omega \right)\right]}^{1/2}$$

Figure 5 shows the dielectric function of Cr^2+^:ZnSe NWs of different sizes. The black, pink, and blue curves represent Cr^2+^:ZnSe NWs with diameters of 8.5 Å, 14.2 Å, and 19.8 Å, respectively. From the real part of the dielectric function, we can see that the permittivity is 1.16, 1.39, and 1.74 for NW-1, NW-2, and NW-3, respectively. The permittivity for all Cr^2+^:ZnSe NW are much smaller than the bulk Cr^2+^:ZnSe (9.41) [34], and monotonically decreases as the diameter of the NWs decreases. And, more importantly, the peaks of $\varepsilon_{1}\left(\omega \right)$ are all positive and dielectric in the whole energy range, while it is negative in the range above 5.5 eV [34]. It means that the Cr^2+^:ZnSe bulk material may transform from metallic to dielectric property above 5.5 eV in the process of becoming NW. For the imaginary part, the curve shows a steep rise at 3.58 eV, 2.92 eV and 2.61 eV for NW-1, NW-2 and NW-3, respectively, which is due to the electron inter-band transition from CBM to VBM. As the band gap increases, the dielectric peak shifts toward higher energy; thus, the absorption edge is blue-shifted. Moreover, compared to pure ZnSe NWs, new peaks appear in the range of 0.7~1 eV are observed, which would lead to additional absorptions in the infrared range.

Figure 6 shows the absorption spectrum of Cr^2+^:ZnSe NWs of different sizes, especially for the infrared range. Absorption peaks at 1.048 eV (1184 nm), 0.862 eV (1440 nm) and 0.719 eV (1726 nm) for NW-1, NW-2, NW-3 are observed, exhibiting a significant blue shift as the NWs diameter decreases. In addition, the absorption of the NWs with the same Cr^2+^ concentration also moved to a shorter wavelength compared to the bulk Cr^2+^:ZnSe [33]. We believe that these absorptions are related to the electron intra-band transitions between the splitting energy level of Cr(d) e and $t_{2}$ orbital. Related mid-infrared luminescence would be generated under corresponding excitation and blue-shifted, which can be verified by our experimental results [35]. The results demonstrate that nano-crystallization is an efficient way to realize the wavelength turnability of laser crystals. 

## 4. Discussion

From the band structure and DOS results of Cr^2+^:ZnSe NWs, it is clear that Cr(d) impurity bands mainly composed of Cr(d) electron orbitals are introduced into the bandgap. The Cr(d) orbitals are divided into two groups, located above and below the Fermi energy level. For d electron orbitals, the nature of the metal ion itself, the oxidation state of the metal, the distribution of ligand ions around the central metal ion, and the nature of the ligand ion around the central metal ion are the main factors that affect the orbital-splitting. The host sphalerite ZnSe is a tetrahedral compound. When Cr^2+^ replaces the Zn^2+^ ion in ZnSe, it occupies the center of the tetrahedron. Due to the action of the ligand electrostatic field, the d orbital splits into higher-energy $t_{2}$ orbitals and lower-energy e orbitals, and the d–d electron transitions are weakly allowed. We inferred that the newly appeared absorption peaks in the infrared range in Cr^2+^:ZnSe bulk and NWs are attributed to the electron intra-band transitions between the splitting energy level of Cr(d) e and $t_{2}$ orbital. To verify this inference, we designed a transition energy statistical algorithm [30,36]. The five Cr-d electron bands are recovered from the band-structure, as displayed in Figure 7a. Cr^2+^ ion has 4 d electrons, occupies the lower e orbital. For the absorption process, electrons at the e orbital absorb just the right amount of energy and jump up to a higher $t_{2}$ orbital, as shown in red arrows. For the luminescence process, electrons at the excited state first spontaneously relax to the metastable level, and then jump back from metastable level to the ground state through radiative transition, as shown in blue arrows. Figure 7b shows the numerical statistics of Cr(d) electron transition energy, shown in the form of a transition energy density distribution. The absorption range is from 0.63 eV to 1.24 eV, 0.57 eV to 1.00 eV and 0.51 eV to 0.90 eV for NW-1, NW-2 and NW-3, respectively, showing a significant blue shift. The results basically match the absorption spectra calculated from VASP. Using the same statistical algorithm, the luminescence would be generated in the range of 0.77 eV~0.63 eV, 0.63 eV~0.57 eV, and 0.73 eV~0.51 eV for NW-1, NW-2 and NW-3, respectively, exhibiting a blue shift as the size of NWs decreases. As it’s a transition energy statistic, the absorption range is wider than the luminescence, which is also reasonable compared to the experimental observation. The disadvantage is that the position of the absorption peak is not accurately predicted because the electron transition probability is treated as equally, and the algorithm model needs a further optimization.

As we have presented, the Cr^2+^:ZnSe NWs show distinct electronic and optical properties and demonstrate a significant size effect. To explore the origin of the property changes, quantum confinement effects and quantum size effects are usually considered [37]. Generally, when one dimension of the nanocrystal is close to the de Broglie wavelength of the electrons ($\lambda_{e}=h/{\left(2m_{e}^{*}kT\right)}^{1/2}$) and holes ($\lambda_{h}=h/{\left(2m_{h}^{*}kT\right)}^{1/2}$) in the semiconductor, quantum confinement effects cannot be omitted. Here, h is the Planck constant, k is the Boltzmann constant, *T* is the Degree Kelvin, $m_{e}^{*}$ and $m_{h}^{*}$ are the effective mass of the electron and hole. In this work, the $\lambda_{e}$ and $\lambda_{h}$ are 7~21 nm (see Table 2), while the Cr^2+^:ZnSe NWs sizes are 1~2 nm, which is much smaller than the $\lambda_{e}$ and $\lambda_{h}$. Obviously, the quantum confinement effects are instrumental in the present work.

Quantum size effects start to make a valuable contribution when the size of semiconductor nanostructures shrinks to less than the Bohr radius of the exciton. The radius of the exciton is given by $a_{ex}=\frac{\hbar^{2}}{m^{*}\left(e^{2}/\varepsilon \right)}=\frac{\varepsilon m_{e}a_{0}}{m^{*}}$ in terms of Bohr radius ($a_{0}=\frac{\hbar^{2}}{m_{e}e^{2}}$), where the reduced effective mass $m^{*}$ is given by $\;\frac{1}{m^{*}}=\frac{1}{m_{e}^{*}}+\frac{1}{m_{h}^{*}}$, and the static crystal dielectric constant ε can be estimated from the arithmetic mean value of the diagonal components of the real part $\varepsilon_{1}$ of the dielectric tensor at ω = 0 Hz,$\;\varepsilon =\frac{1}{3}\times \left(\varepsilon_{1}^{\mathrm{xx}}\left(0\right)+\varepsilon_{1}^{\mathrm{yy}}\left(0\right)+\varepsilon_{1}^{\mathrm{zz}}\left(0\right)\right)$. Here, the calculated exciton’s Bohr radius are 173 Å, 6.19 Å, 5.23 Å and 7.86 Å for Cr^2+^:ZnSe bulk, NW-1, NW-2 and NW-3, respectively. The radius of the exciton’s Bohr are of the same order as the size of the Cr^2+^:ZnSe NWs. So, it is believed that quantum confinement effects also play an important role in these NWs. 

During the optical transitions, the electrons excited into the conduction band and the holes in the valence band will form a bound state, called an exciton, due to Coulomb interactions. Excitonic effects have an important impact on the physical processes and optical properties in semiconductors, such as optical absorption, luminescence, lasing, and optical nonlinearity. Based on the effective masses of electrons and holes at the CBM and VBM, the exciton binding energy can be obtained as [31]:(5)$$E_{ex}=\frac{e^{2}}{2\varepsilon a_{ex}}=\left(\frac{m^{*}}{m_{e}}\right)\cdot \left(\frac{1}{\varepsilon^{2}}\right)\cdot 13.6\;eV$$

The calculate exciton binding energy are 0.004 eV, 1.00 eV, 0.99 eV and 0.53 eV for Cr^2+^:ZnSe bulk, NW-1, NW-2 and NW-3, respectively. A larger exciton binding energy presents a stronger electron–hole interaction. Here, the excitons binding energy of Cr^2+^:ZnSe NWs is considerably higher than that of the bulk, indicating a presence of strongly bound excitons in Cr^2+^:ZnSe NWs. This enhanced exciton binding energy in NWs suggests that the excitons may bind to possible defects in the nanocrystals and act as recombination centers, thus promoting the luminescence efficiency. They are very beneficial to the fabrication of optoelectronic devices with improved performance. 

## 5. Conclusions

In summary, the electrical and optical properties of Cr^2+^:ZnSe NWs of different sizes with Cr^2+^ at different sites were calculated. Based on the defect formation energy results, we reveal that the Cr^2+^ prefers residing at the surface of ZnSe NWs. As the size of the NWs decreases, a considerable short-wave-length shift in the fundamental absorption is observed. The absorption in the visible range is related to the electron internal band transitions from VBM to CBM, and the absorption peak that appeared in the infrared range is ascribed to the electron intra-band transitions between the splitting energy level of Cr(d) e and $t_{2}$ orbital, and related mid-infrared luminescence would be generated under corresponding excitation and also blue-shifted. The blue-shift trend is confirmed by the numerical statistics of Cr(d) electron transition energy. Quantum confinement effects and quantum size effects play important roles in the new electrical and optical properties of Cr^2+^:ZnSe NWs. Overall, the electrical and optical properties of Cr^2+^:ZnSe NWs are dependent on the nanostructure size, which suggests that Cr^2+^:ZnSe NWs may be used in promising optoelectronic applications.

However, in the physical fabrication of the nanomaterials, the introduction of defects is inevitable and the morphology of the nanostructures is strongly dependent on the fabrication technology. A more accurate model should consider nanocrystal morphology, nanocrystal composition, and dopant surface coverage, etc. Our nearest future intension is to develop the model of these nanocrystals and study the interaction between defects and impurity ions furtherly. 

## Figures and Tables

**Figure 1 nanomaterials-13-00369-f001:**
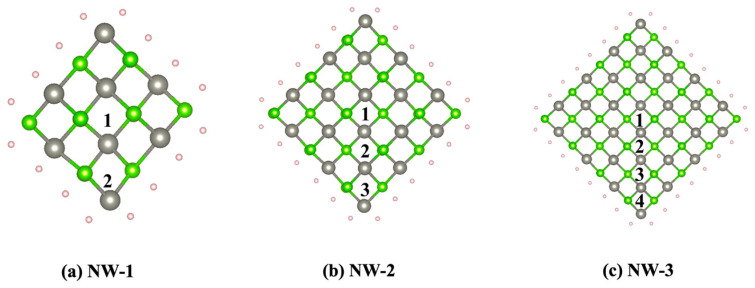
Top view of ZnSe NWs along the [001] direction with different diameters: (**a**) NW-1(8.5 Å), (**b**) NW-2(14.2 Å), (**c**) NW-3(19.8 Å). Green and grey atoms are Se and Zn atoms, red and blue atoms are H1.5 and H0.5 atoms, respectively.

**Figure 2 nanomaterials-13-00369-f002:**
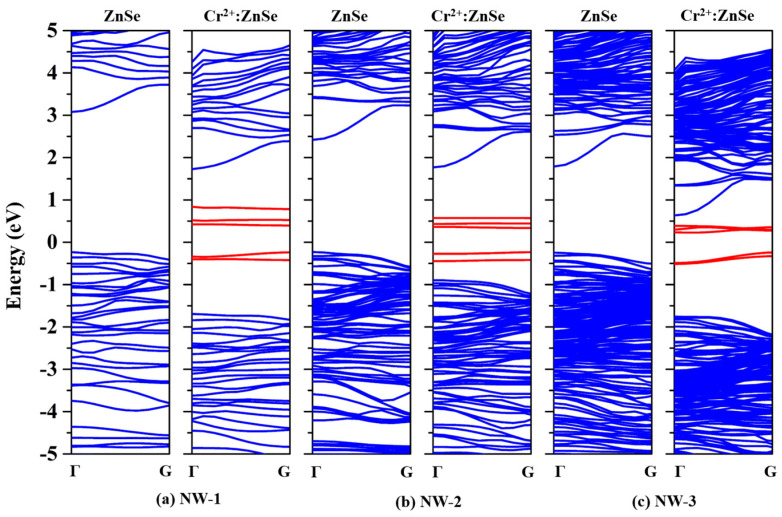
The band structure of ZnSe and Cr^2+^:ZnSe NWs in different sizes. The lines in blue present the intrinsic bands of ZnSe NWs, and the red lines present the impurity bands introduced by Cr atom doping.

**Figure 3 nanomaterials-13-00369-f003:**
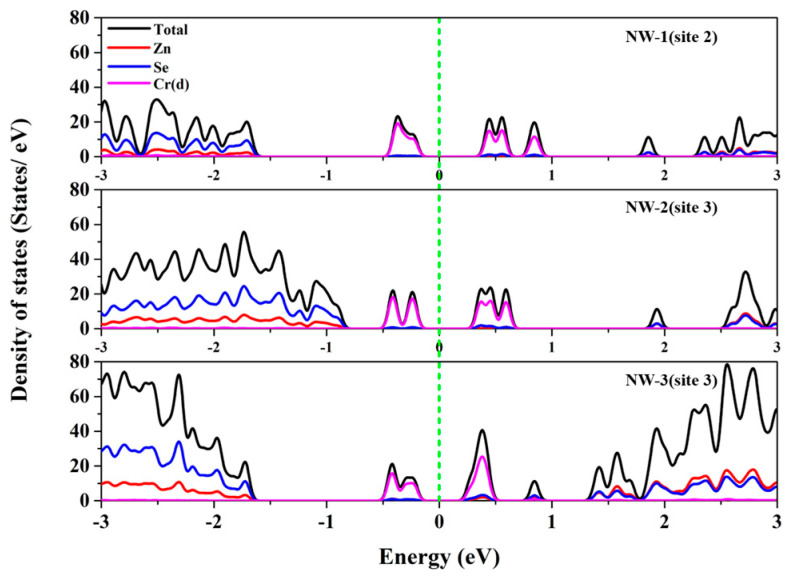
The DOS of Cr^2+^:ZnSe NWs in different sizes.

**Figure 4 nanomaterials-13-00369-f004:**
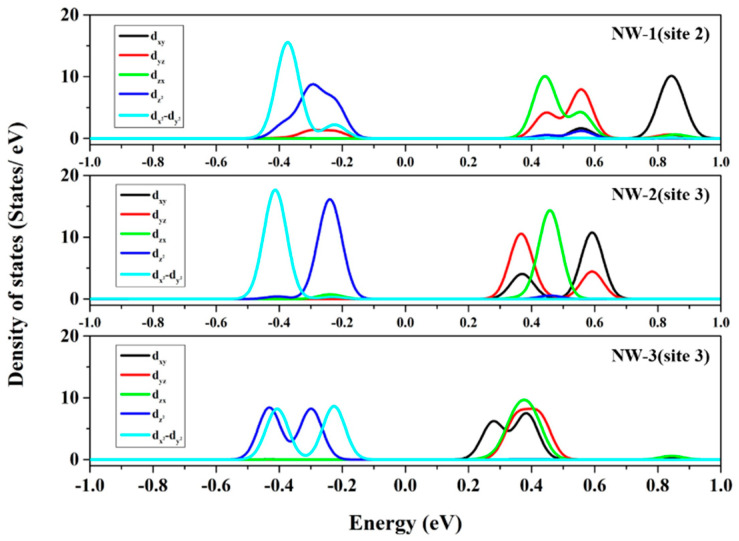
PDOS of Cr^2+^:ZnSe NWs of different sizes.

**Figure 5 nanomaterials-13-00369-f005:**
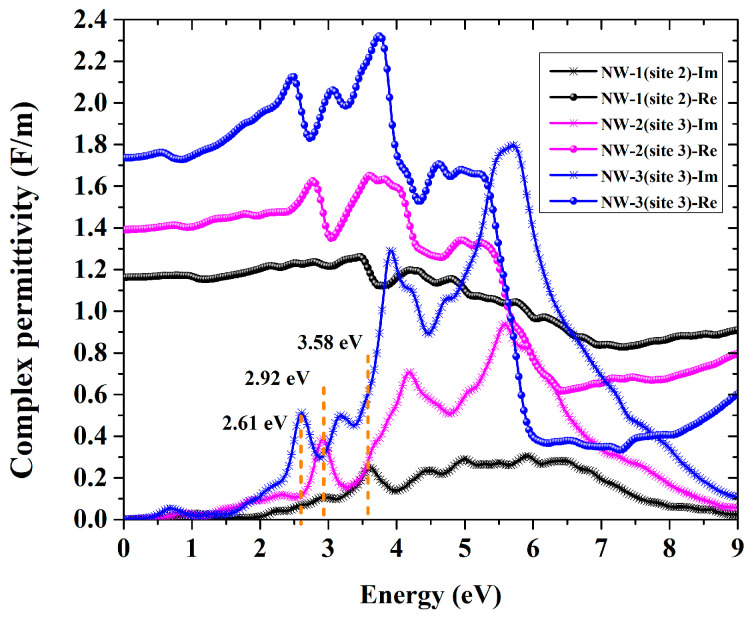
Dielectric function of Cr^2+^:ZnSe NWs of different sizes.

**Figure 6 nanomaterials-13-00369-f006:**
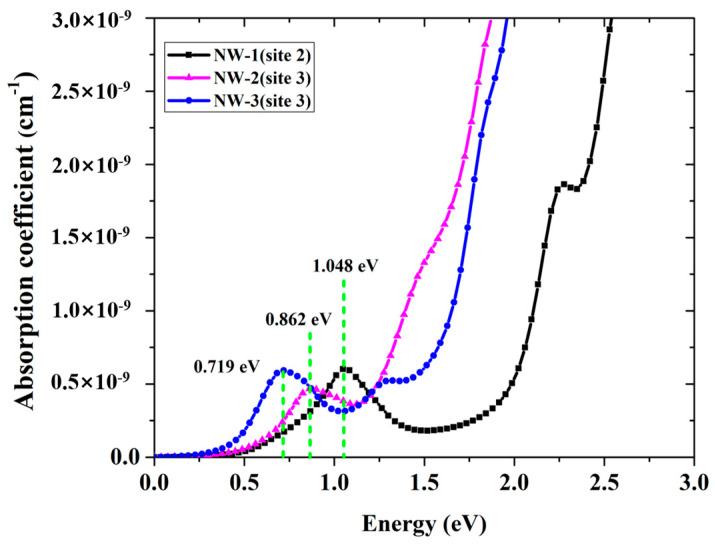
Absorption spectrum of Cr^2+^:ZnSe NWs of different sizes.

**Figure 7 nanomaterials-13-00369-f007:**
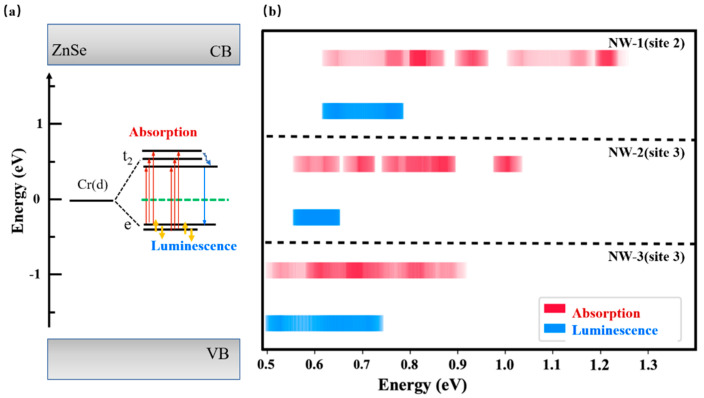
The absorption and luminescence calculation scheme (**a**) and numerical statistics of Cr(d) electrons transition energy density distribution of Cr^2+^:ZnSe NWs (**b**).

**Table 1 nanomaterials-13-00369-t001:** The total energy and defect formation energy of Cr^2+^:ZnSe NWs in different sizes.

Models	Total Energy (eV)	Defect Formation Energy (eV)
NW-1(Zn8Se8)	−87.412	/
NW-1(site 1)	−92.257	2.265
NW-1(site 2)	−93.066	1.456
NW-2(Zn18Se18)	−168.793	/
NW-2(site 1)	−174.461	1.442
NW-2(site 2)	−174.230	1.673
NW-2(site 3)	−174.905	0.998
NW-3(Zn32Se32)	−272.860	/
NW-3(site 1)	−278.982	0.988
NW-3(site 2)	−279.691	0.279
NW-3(site 3)	−279.938	0.030
NW-3(site 4)	−279.572	0.398

**Table 2 nanomaterials-13-00369-t002:** The bandgap, effective mass values and de Broglie wavelength for Cr^2+^:ZnSe NWs and bulk.

Models	Size (Å)	Bandgap (eV)	$\mathrm{Effective}\;\mathrm{Mass}\;(m_{0})$		de Broglie Wavelength (nm)	
			$m_{e}^{*}$	$m_{h}^{*}$	$\lambda_{e}$	$\lambda_{h}$
NW-1 (site 2)	8.5	3.42	0.34	−0.14	13	20
NW-2 (site 3)	14.2	2.67	1.17	−0.16	7	19
NW-3 (site 3)	19.8	2.39	1.19	0.13	7	21
bulk [30]	/	2.14	0.03	−0.69	44	9

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
