# Peer review of "Size Effect of Electrical and Optical Properties in Cr2+:ZnSe Nanowires"

_nanomaterials, 2023, doi:10.3390/nano13020369_

Round 1

Reviewer 1 Report

This project aims to explore the influence of nanocrystalline size on the electrical and optical properties of Cr2+:ZnSe nanowires  The Authors showed that as the size of the  NWs being decreased, a considerable short-wave-length shift of the fundamental absorption and luminescence are observed.

The Authors have presented sufficient data. The appropriate tables and figures have been provided. The article is easy to read and logically structured.  The methods are adequately described. The conclusions are consistent with the presented evidence and arguments.

There are only some  comments in the reviewer's opinion that should be taken under consideration by the Authors:

1.       Please choose the type of your manuscript as Article

2.       Please add the limitation of your study

3.       Please separate section results from the discussion section

4.       Please add :

a)       Author contributions

b)      Funding:

c)       Institutional Review Board Statement:  

d)      Informed Consent Statement:

e)      Data Availability Statement:

f)        Conflicts of Interest:

5.      . Please prepare references according to the Author’s guidelines

Author Response

Thank you for your comments. We really appreciate it. 

  1. The type of the Manuscript is Article.
  2. The limitation of the study is added in the conclusion section.
  3. The discussion section is separated from the results section.
  4. The Author contributions, Funding, Data Availability Statement and Conflicts of Interest are added at the end of the Manuscript.
  5. The references are prepared according to the Author’s guidelines.

Reviewer 2 Report

Show a title and the unit of horizontal axis of figure 7 clearly in a figure.

Figure. 7 ; Dispersion of Absorption is bigger than a result of Luminescence. Show this reason.

Author Response

Thanks for your comment and we appreciate it.

  1. The title and unit of the horizontal axis are added in Fig.7.
  2. As it’s a statistic of the transition energy, the absorption range is wider than the luminescence, a detailed description of Fig.7 is added. 

Reviewer 3 Report

The article deals with the investigation of the effect of nanocrystal size on the electrical and optical properties of nanowires. With the development and new application possibilities of nanostructures, there is room for the application of nanowires. It is predicted that the development of nanostructures will be crucial for many fields such as medicine, telecommunications, transportation equipment, space applications, etc. The development of nanostructures is essential and many devices will not be able to exist without these structures. Therefore, I consider this issue to be necessary and necessary to solve. The issue is highly topical and will certainly be of interest to readers and application engineers from practice.

The introduction contains the basic facts of the current situation and the justification of the need to solve this issue. The methods for solving the problem are appropriately chosen and the results are clearly presented. However, the article still needs to revise some parts and remove formal deficiencies.

Comments:

There are many formal formatting errors in the article such as:

(line 24: "areas[1, 2]." - missing space

line 26: "features[3]." - space is missing

and this is how it is throughout the article. It all needs to be fixed).

Why are the references in bold style on lines 73, 74 and 75?

All equations are written in a large font compared to the rest of the text. It looks terrible.

Also, the symbols are very large, for example on line 99, 100, 101, 102, etc. This needs to be corrected throughout the article.

The equations are not numbered at all.

In table 2: Effective mass has no units.

The title of Figure 7 is misplaced. See the template how it should be right.

Images should be in a uniform style. Look at picture 5 how big the text is and picture 4 is different. It needs to be unified in all images. Now it looks bad.

On line 221 is "dielectric tensor at omega" - if omega is a matrix or vector, it should be in bold style. If not, then it should be like italic.

There is no discussion in the article and the conclusion of the article is extremely brief. A scientific article must have a separate chapter where there is a detailed discussion of individual research outputs and a critical evaluation of the results achieved. In conclusion, the own contribution and novelties of the article must be emphasized. There must also be a plan for future research in this area.

Links to articles in this journal are missing from the references. In this journal there are articles with similar issues. Why are they not cited?

Author Response

Thanks for your comment and we appreciate it.

  1. The formal formatting errors are corrected. 
  2. We have corrected the references in bold style.
  3. The equations and symbols are adjusted to the suitable size and all the equations are numbered.
  4. The Effective mass in Table 2 is in the unit of the electron mass.
  5. The title of Figure 7 is replaced.  
  6. All the images are now in a uniform style, with the same size. 
  7. The dielectric tensor at omega is a matrix, and we have corrected it in bold style.
  8. The discussion section is added. In the conclusion, the own contribution and novelties of the article are emphasized and a plan for future research is added.
  9. Articles in this journal with similar issues are cited.

Reviewer 4 Report

This paper evaluated the chemical properties of Cr2+:ZnSe nanoparticles. Please update the Methods section to include detailed descriptions of the materials and methods used in this study. The authors include a brief description of the calculations and models developed, but details regarding this analysis should be included in the manuscript.

-Please include further description within Fig. 7 legend and format appropriately.

-Please correct the typographical issues throughout- example (line 181) - fragment sentence 'Which...'

Author Response

Thank you for your comments. We really appreciate it. 

  1. A further description of Fig. 7 is added.
  2.  The typographical issues are corrected.